# Parent–Child Interaction Therapy Supports Healthy Eating Behavior in Child Welfare-Involved Children

**DOI:** 10.3390/ijerph191710535

**Published:** 2022-08-24

**Authors:** Emma R. Lyons, Akhila K. Nekkanti, Beverly W. Funderburk, Elizabeth A. Skowron

**Affiliations:** 1Pediatric Mental Health Institute, Children’s Hospital Colorado, Aurora, CO 80045, USA; 2Center for Innovation and Research on Choice-Filled Lives, Choice-Filled Lives, Inc., Atlanta, GA 30305, USA; 3Department of Developmental and Behavioral Pediatrics, Oklahoma Health Sciences Center, Oklahoma City, OK 73104, USA; 4Department of Psychology, University of Oregon, Eugene, OR 97403, USA

**Keywords:** PCIT, parenting, child eating behaviors, obesity prevention, food insecurity, adverse childhood experiences

## Abstract

Objective: We tested the efficacy of standard Parent–Child Interaction Therapy (PCIT), a live-coached, behavioral parent-training program, for modifying problematic eating behaviors in a larger effectiveness trial of PCIT for children involved in the child welfare system. Method: Children ages 3–7 years and their parents were randomly assigned to PCIT intervention (*n* = 120) or services as the usual control (SAU; *n* = 84) groups in a randomized clinical trial. Children’s eating behaviors were assessed pre- and post-intervention via the Child Eating Behaviors Questionnaire (CEBQ). Intention-to-treat analyses were conducted, followed by per-protocol analyses, on treatment-engaging families only. Results: PCIT led to reductions in child welfare-involved children’s food responsiveness, speed of food consumption, and tendency to engage in emotional overeating relative to children in the services-as-usual control condition. Standard PCIT may be an effective intervention to promote healthy child eating behaviors in families involved with child welfare, even when food-related behaviors are not directly targeted by the intervention. Public Health Significance: This clinical trial provides evidence that child welfare-involved children who received PCIT experienced significant reductions in maladaptive eating-related behaviors, namely food responsiveness, emotional overeating, and speed of eating. These findings were observed in relation to children in a comparison control group who had access to child welfare services-as-usual.

## 1. Introduction

Childhood abuse and neglect remains a significant public health concern in the United States, with approximately 615,000 cases of maltreatment occurring in 2020 alone [1]. Along with maltreatment, child welfare-involved children are more likely to experience poverty, family instability and conflict, food insecurity, and caregiver substance abuse and/or mental health concerns [2], all of which place children at an increased risk for long-lasting emotional, behavioral, and physical health problems that can last into adulthood [3,4]. Exposure to adverse childhood experiences (ACES) also has a profound impact on a child’s developing biology and can precipitate alterations in neural, cardiovascular, neuroendocrine, and immune functioning [5] and lead to negative health-related behaviors. Childhood maltreatment exposure has been linked to higher subsequent rates of high-fat eating, cigarette smoking, alcohol and drug consumption, physical inactivity, [6] and other unhealthy eating behaviors [7].

Indeed, ACE-exposed, food-insecure children are at a higher risk for obesity and later eating disorders [7,8,9,10,11]. Families experiencing food insecurity often depend on nutrient-dense, low-cost, processed food in order to extend their available food supply and money [12], which could contribute to obesity due to the overconsumption of calorically dense foods, or to being underweight due to food scarcity [13,14]. Biopsychosocial factors related to ACEs also contribute to maladaptive parent–child behavioral exchanges during mealtimes [15], resulting in children’s dysregulated eating patterns [5,16]. For instance, parents may pressure their child to eat to avoid wasting available food [17] or to restrict their children’s food intake to ensure it lasts longer [18]. Food insecurity leaves children at risk for emotional overeating when food is available [19] and greater food responsiveness, which involves eating beyond satiety in response to external food cues (i.e., sight or smell). Emotional overeating and food responsiveness are both associated with a larger caloric intake at meals and faster rates of eating [20] and contribute to the risk for childhood overweight status [12,21,22]. 

Parental feeding practices during mealtimes are key factors that influence how children eat, the quality of their child’s diet, and overall healthy growth [15,23]. Parenting strategies that support children’s age-appropriate autonomy, such as making food available and accessible and modeling healthy eating behavior, tend to promote their children’s healthy eating [24]. Similarly, parenting that is warm, responsive, and authoritative (e.g., parenting practices that include attentiveness to child needs coupled with appropriate support for autonomy; [25,26,27]) has been shown to promote healthy eating practices in children and buffer against the risk for childhood obesity, poor diet, and poor growth [28,29]. Responsive parents provide a variety of developmentally appropriate foods, and the child decides the quantity and type of food they wish to eat [23,30], free from pressure or coercion. Responsive feeding practices ultimately promote the development of a child’s self-regulation skills and related food intake, thereby preventing excess weight gain [28,29,31]. In contrast, parenting strategies that involve high levels of control (e.g., restricting food intake, pressuring the child to eat, and using food as a reward for eating) are associated with children’s unhealthy food consumption, food refusal, and other fussy eating behaviors [32]. Coercive and controlling parenting around eating interferes with a child’s ability to sense their own hunger and satiety cues, and perpetuates emotional overeating and food responsiveness [33]. As child welfare-involved parents experience many contextual stressors (e.g., poverty, food insecurity, housing instability, and household chaos) that leave them vulnerable to using coercive, nonresponsive feeding behaviors [34,35,36] parenting behavior and the parent–child relationship quality are important targets for fostering healthy eating in the context of psychosocial risk and adversity.

Interventions exist that target children’s problematic eating behaviors, most often in the context of obesity prevention; however, their effects on promoting long-term behavior changes and improving health outcomes are mixed, depending on their format and the ages of the children involved [37,38]. Several meta-analyses have shown that family-based interventions are more effective than individual treatments for reducing obesity in children [39,40]. Most recently, Melis Yavuz and colleagues’ (2015) meta-analysis reported that the largest effects on children’s weight-related outcomes post-treatment were achieved by family-based interventions that focused on parenting practices, with the retention of gains at the long-term follow-up observed in family-based interventions that targeted preschool-aged children [41,42,43,44]. Family interventions that focused specifically on changing both general parenting skills and mealtime-specific parenting behaviors during joint parent–child sessions were found to be superior to interventions that focused only on mealtime-specific behaviors and which utilized didactic formats [38]. Further evidence from a recent randomized control trial showed that a parent-only intervention comprised of evidence-based parenting practices outperformed a standard lifestyle-focused treatment for reducing BMI scores in obese children [37]. However, to our knowledge, no studies have examined the effectiveness of parenting interventions for supporting children’s healthy eating outcomes in child welfare-involved families. Among these high-needs families where food instability, insecurity, and high stress levels are prevalent, parenting interventions that strengthen responsive parenting and that support parent–child relational quality may also function to promote healthy eating practices in children. 

Parent–Child Interaction Therapy (PCIT) is an evidence-based family intervention that teaches parents to adopt a warm, responsive, authoritative parenting style while decreasing coercive, authoritarian behaviors, the effects of which may extend into mealtimes with their children. Originally designed to address behavior concerns in young children aged 2–7 years, PCIT is rooted in social learning and attachment theory and therefore emphasizes the parent–child relationship as a key mechanism for child behavior change. PCIT has been shown to decrease harsh, controlling, and coercive parenting while promoting positive parenting behaviors that include responsiveness and support for a child’s autonomy [45,46]. Strong evidence also supports PCIT’s ability to lower the risk for child abuse and reduce aversive discipline strategies in families in the child welfare system [47,48,49,50]. Due to its focus on promoting warm, responsive parenting, PCIT may indeed be an appropriate intervention to address children’s more challenging eating behaviors. 

## 2. Current Study

This study employed a randomized clinical trial design to investigate whether the standard PCIT intervention—which focuses on promoting positive and responsive parenting and reducing harsh, aversive parenting—would also demonstrate beneficial effects on children’s eating behavior styles in a sample of child welfare-involved families. We hypothesized that children in the PCIT condition would display healthier eating behavior styles post-treatment, as compared to children in the CW Services-as-Usual (SAU) condition. Given that standard PCIT has demonstrated strong, positive effects on child welfare parent and child outcomes [48,49,50] and has demonstrated effectiveness with racial/ethnically diverse families [51,52] and for various presenting concerns over and above oppositional behavior (e.g., selective mutism: [53]; chronic illnesses: [54]; and foster care children; [55]) we reasoned that its beneficial effects would extend to child eating behaviors, even though feeding practices were not targeted.

## 3. Method

### 3.1. Participants and Design

A sample of 204 young children and their parents were recruited from the Department of Human Services (DHS) by their child welfare and self-sufficiency caseworkers and screened by research staff for study inclusion. Children 3–7 years old (M = 4.76, SD = 1.40 years) were recruited into the study, and 45.1% were girls. The majority of children (69.1%) were exposed to 3 or more Adverse Childhood Experiences (ACES) before entering the study. Children’s participating parents ranged in age from 18 to 64 (M = 32.3, SD = 6.4) years old and were predominantly mothers (88%) and biological parents (98%) of the children. Based on the 2020 U.S. DHHS guidelines, 78.5% of family households were living below the federal poverty line. Considering all ethnicities (i.e., summing up to greater than 100%), 19.6% of children in the total sample identified as Native American/Alaskan Aleut, 15.2% Hispanic, 8.8% Black/African American, 2.9% Asian/Asian American, 3.4% Pacific Islander, and 93.1% White/European American. The socio-demographics are presented in Table 1, organized by condition. Randomization tests confirmed that no significant differences were observed across the conditions on the children’s sociodemographics, ACEs scores, or the outcomes (i.e., children’s eating behaviors).

### 3.2. Procedures

This study was registered with clinical trials.gov (NCT02684903). Families were referred to the study through DHS beginning in the spring, 2016 through June 2019. First, DHS staff members identified eligible families from their database. Following this pre-selection process within DHS, a core member of the research team contacted each family to invite them to participate in the study and further screened them for eligibility criteria: Participating parents were at least 18+ years old at study entry and the participating child’s biological or custodial parent; no parent or caregiver in the home was a documented child sexual abuse perpetrator per child welfare records; and the parent and child spoke sufficient English to engage in the assessment. Families were free to participate or decline and were informed that they would be randomized to either the PCIT intervention or continue receiving services as usual following the completion of their baseline assessment visits. Families were block randomized to the standard length PCIT intervention or DHS services-as-usual (SAU) control condition by child sex and age using a 1.5:1 allocation ratio (intervention to control condition, respectively). Baseline assessment visits were initiated for 228 families, and of those, *n* = 23 did not complete the baseline assessment and 1 withdrew from the study, resulting in a sample of *n* = 204 families enrolled and randomized to the condition. Written informed consent for participation was obtained from parents, and in cases where DHS maintained legal custody of the child, the child’s child welfare caseworker also provided written consent. Parent–child dyads were compensated for attending assessments pre- and post-intervention; reimbursed for transportation costs; and received refreshments, rest breaks, and childcare for their nonparticipating children. Participating children also received a small prize. Post-intervention assessments were completed on the same timeline for the intervention and control conditions at M = 7.8 (SD = 2.3) months post-study entry. Further details on the study procedures are available by Nekkanti et al. (2020) [56].

### 3.3. Intervention: Parent–Child Interaction Therapy (PCIT)

PCIT is an intensive, behavioral parent training program that employs a live-skills coaching approach to support parents in real time as they practice new parenting skills in sessions together with their child to support safe, nurturing, effective parenting [46,57]. PCIT starts with Child-Directed Interactions (CDI), in which parents learn first to follow their child’s lead in the play, use skills intended to strengthen the warmth in the relationship, support their child’s developing autonomy, and refrain from harsh, aversive parenting. Next in Parent-Directed Interactions (PDI), parents learn and practice child behavior management and time-out procedures. In PDI, parents learn to give clear, direct, developmentally tailored commands, follow through with each, then transition back into the use of CDI skills. Parents learn safe, effective alternatives to the use of harsh or erratic discipline techniques. Children are introduced to the PDI phase sessions with an explanation that they will practice “minding and listening” or doing what their parent says and are provided a demonstration to ensure their understanding.

*Intervention Integrity.* PCIT was delivered using a manualized protocol [57], and therapists completed fidelity checklists following each session. The standard length PCIT protocol consisted of an intake, up to 9 CDI sessions and 11 PDI sessions, with study families completing an average of 7.2 (SD = 3.2) CDI sessions and 5.6 (SD = 5.0) PDI sessions. PCIT sessions were delivered by six Ph.D. student therapists, a licensed social worker, and a licensed psychologist), whose training and supervision conformed to the PCIT international standards for observed practice and intervention fidelity criteria. University of Oklahoma Health Sciences Center global and regional trainers provided weekly remote consultations and live supervision of the intervention delivery. All sessions were also video-recorded to enable independent ratings of therapist fidelity to the PCIT protocol. Trained fidelity raters blind to the session number coded 20% of the PCIT sessions (i.e., 203 out of 1013 sessions), with the results showing high therapist adherence (95%) to the PCIT treatment manual protocol. 

### 3.4. Services-as-Usual Control (SAU)

The no-treatment control group condition represented an ecologically valid, ethical comparison condition in which families retained access to various services provided by DHS child welfare and self-sufficiency units. Thus, families in the SAU control condition may have received services including in-home family visitation services, respite childcare, other individual child counseling, parent education training, and/or individual adult therapy, including treatment for substance use disorders. 

## 4. Measures

The Child Eating Behavior Questionnaire (CEBQ; [21,58]. The primary outcomes in this report were child eating behaviors, assessed via parent-reporting on the CEBQ. The CEBQ is a 35-item parent-report survey of eating styles in children ages 2–12 years. Parents complete items using a 5-point scale from 1 (never) to 5 (always). The CEBQ contains eight scales: Food responsiveness (4 items), Emotional overeating (4 items), Satiety responsiveness (5 items), Slowness in eating (4 items), Emotional undereating (4 items), Fussiness (7 items), Enjoyment of food (4 items), and Desire to drink (3 items). 

The Food Responsiveness (FR) scale reflects tendencies to eat in response to environmental cues. The Emotional Overeating (EO) scale measures increased eating in response to negative emotions, and the scores positively correlate with child BMI [59]. Children who experience food insecurity and heightened family stress display greater food responsiveness and emotional overeating than children in food-secure, lower-stress families [12]. The Emotional Undereating (EU) scale assesses decreased eating in response to emotions such as anger or anxiety/stress. Satiety Responsiveness (SR) reflects how well a child is able to reduce food intake after eating. The Slowness in Eating (SE) scale measures the speed of eating during meals and the gradual reduction in one’s interest in the meal. The eating rate has been shown to increase in overweight and obese children [59] and as children age [60,61]. The Food Fussiness scale measures the tendency to reject a variety of familiar and novel foods and limit the variety of foods consumed. The Enjoyment of Food (EF) scale assesses a child’s general interest in food, and the Desire to Drink scale taps the desire of children to have drinks available to carry with them. Internal consistency of the CEBQ subscales is good, with the Cronbach’s alphas in the current sample as follows: Food responsiveness 0.82, Emotional overeating 0.86, Satiety responsiveness 0.79, Slowness in eating 0.81, Emotional undereating 0.74, Food Fussiness 0.93, Enjoyment of food 0.89, and Desire to drink 0.82. The construct validity of the CEBQ is strong: CEBQ scores have been shown to correspond with directly observed measures of eating behaviors in children [21]. Moreover, studies have shown children’s body mass index (BMI) z-scores (i.e., controlling for age and biological sex) are positively associated with scores on the CEBQ food approach scales (Emotional Overeating, Food Responsiveness, and Enjoyment of Eating) and negatively associated with scores on the food avoidant scales (Satiety Responsiveness, Slowness in Eating, Emotional Undereating, and Food Fussiness; [62]).

Sociodemographics. Parents completed other survey measures, such as family sociodemographics, child characteristics, and their child’s exposure to early adversity on the Adverse Childhood Experiences Scale (ACES; [3]), with a research assistant using an interview format due to the wide variability in the parents’ reading levels.

### Analytic Plan

Intention-to-treat (ITT) analyses were conducted with all randomized participants (*n_PCIT_* = 120, *n_Control_ =* 84) on multiple imputed data. Next, per-protocol analyses were conducted with all children in the PCIT condition who engaged in the treatment (*n_PCIT_* = 79) and all those in the SAU control group (*n_Control_ =* 84). We applied multiple regression on the multiply imputed data to test the effect of PCIT on child eating outcomes for both the ITT analysis (i.e., with the groups exactly as randomized; *n* = 204) and per-protocol analyses (i.e., with treatment engagers versus the control group participants only; *n* = 163). Missing variables at pre- and post-treatment were imputed using predictive mean matching in the ‘mice’ package in R (Version 3.3.0; [63]). Parameters for the analyses were estimated in each of 40 imputed datasets [64] and pooled using Rubin’s rules. Each analysis controlled for the pre-test values of the post-test dependent variables. The effect sizes (i.e., Cohen’s *f*^2^) were calculated using guidelines provided by Selya et al., 2012 [65] and represented the effect of treatment on the child outcome variable after accounting for the baseline values. 

## 5. Results

Missing data did not significantly vary by condition across any variables. Post-intervention outcomes in the parents are presented elsewhere [66]. In brief, the standard, time-limited PCIT intervention facilitated gains in positive, responsive parenting, and reductions in harsh, aversive parenting, both in the context of free play and during child management interactions, and improved the parents’ inhibitory control and emotion regulation skills relative to the parents in the SAU control condition [66]. 

### Effects of PCIT on Children’s Eating Behavior Outcomes

Table 2 shows the means and standard deviations for each of the CEBQ eating outcome variables at pre- and post-treatment for children in the PCIT intervention group and children in the SAU control group. Table 2 also provides the pooled parameter estimates, effect sizes, and 95% confidence intervals for each ITT effect of PCIT relative to the SAU control and each per-protocol effect of PCIT engagers versus SAU controls. The baseline values of the outcome scores were accounted for in the analyses. Positive values for the unstandardized betas indicate that PCIT children scored higher than the control group children on the outcome, whereas negative estimates indicate that children in the PCIT condition scored lower.

After accounting for pre-treatment CEBQ eating behavior scores, PCIT had significant ITT and per-protocol (PP) effects on reducing child welfare-involved children’s Food Responsiveness at post-treatment (ITT: (*t*(118) = −2.19, *p* = 0.03, *f*^2^ = 0.034; PP: *t*(102) = −2.08, *p* = 0.03, *f*^2^ = 0.038)). The results indicate that PCIT also had a significant effect on increasing child welfare children’s Slowness in Eating following the intervention, according to both the ITT and per-protocol analyses [ITT: (*t*(125) = 2.41, *p* = 0.017, *f*^2^ = 0.04; PP: *t*(99) = 2.34, *p* = 0.02, *f*^2^ = 0.05)]. Small to small/medium effects were observed. Finally, we also observed lower Emotional Overeating scores at post-treatment for children in the PCIT condition versus the control group, though the difference did not reach statistical significance: ITT model: *t*(121) = −1.79, *p* = 0.07, *f*^2^ = 0.02 and PP model: *t*(102) = −1.68, *p* = 0.09, *f*^2^ = 0.03.

Clinical Significance. Table 2 shows Cohen’s *f*^2^ effect sizes and confidence intervals for the effect of PCIT on post-treatment outcomes, after accounting for the effect of pretreatment scores, for both the ITT and per-protocol samples. Cohen’s (1988) [67] guidelines for interpreting the magnitude of the effect sizes indicate that *f*^2^ ≥  0.02 values represent small effects, *f*^2^ ≥  0.15 values reflect medium effects, and *f*^2^ ≥  0.35 values represent large effect sizes. PCIT’s effects on reducing children’s food responsiveness, emotional overeating, and increasing slowness in eating were similar in magnitude, reflecting small effect sizes that were relatively comparable across the ITT and per-protocol models. 

## 6. Discussion

PCIT was originally developed to treat disruptive behavior problems in children and has been successfully applied to reduce the risk for child maltreatment and support positive, nurturing parenting in child welfare-involved families. The findings from this randomized clinical trial provide new evidence that standard length PCIT also improves Food Responsivity, Eating Speed, and Emotional Overeating in a sample of high-risk, ACE-exposed children. Children who received PCIT showed reduced food approach behaviors (e.g., reduced food responsivity or tendency to eat past satiety and emotional overeating) and increased slowness while eating post-treatment, compared to children randomized to the control group. It is important to note that children included in this study did not specifically present with eating concerns, and eating behavior was not the target of PCIT within the larger clinical trial. However, it is well-established that ACE-exposed children who live with food insecurity tend to display greater food approach behaviors (e.g., Food Responsivity and Emotional Overeating) compared to food-secure children [12] and that these behaviors also place children at a higher risk for compromised health outcomes throughout development [21]. Thus, although the children included in this study did not present specifically with eating concerns, several high-risk eating behaviors that have been associated with later concerns appeared to improve within children in the PCIT group. These results suggest that standard PCIT may be a helpful intervention to address problematic food approach behaviors in ACE-exposed children, even when the focus of the intervention is aimed at improving positive parenting skills rather than directly targeting children’s eating. 

Further, these findings are in line with other published intervention studies, which demonstrate that parenting behavior is an important intervention target to improve child eating patterns and reduce the risk for childhood obesity [24,37,38]. With its robust effects on warm, authoritative parenting skills [49,50], standard PCIT may lend itself well to inclusion in comprehensive programs to promote healthy eating and prevent obesity and other eating-related disorders throughout development. Future research efforts are needed to examine whether standard PCIT can help to support the positive outcomes achieved via other existing childhood obesity prevention approaches, including family and school-based physical activity and dietary interventions [68]. How does PCIT confer changes in children’s eating behavior within ACE-exposed, child welfare-involved families? The combination of increasing warmth and connection in the parent–child relationship, along with supporting parental gains in safe, effective child management skills, may set the stage for a more calming home environment [48]. In child welfare-involved families, the most proximal risks for child maltreatment are negative and coercive patterns of parent–child interactions and parents’ lack of knowledge or inappropriate use of discipline [69], which PCIT directly targets with success through its emphasis on positive behavior management skills and consistent discipline (e.g., [70]). Parent–child conflict often sparks from predictable challenges of typical child development—for example, during bedtime, bath time, and mealtimes. By disrupting coercive parent–child exchanges and decreasing controlling parenting strategies across these daily challenging scenarios (e.g., mealtime), PCIT helps parents to promote safety and support their child’s autonomy while effectively managing their behavior. However, further investigation is needed to ascertain whether gains in these parenting skills (a) help children calm, better modulate their eating behaviors, and resist using food to self-soothe and (b) transfer to parenting in the context of mealtimes through the observation and coding of in vivo mealtime parent–child interactions and testing whether specific changes in parenting function as mediators of these outcomes. 

Though child welfare-involved children rarely present to services for eating-related behavior problems, nonetheless, they remain at a higher risk for a host of problematic health outcomes. According to the Neuroimmune Network Hypothesis [5,71], children exposed to early toxic stress remain highly vulnerable to physical and mental health problems across the lifespan (e.g., major depression, type 2 diabetes, and heart disease, e.g., [72,73,74]. These high levels of early life stress are known to impact children’s health both directly, by causing low-grade, chronic inflammation [75,76], and indirectly through pathways that amplify negative health-related behaviors used to cope or self-soothe (e.g., consumption of high-fat and high-sugar diets and other maladaptive eating styles, as well as smoking, alcohol, and drug use). Child welfare services provide critical, tangible resources (e.g., financial support for food, clothing, housing, and transportation) for families in need, which help to blunt the effects of toxic stress on children’s developing neurobiology [77,78]. More research is needed to determine if the beneficial effects PCIT exerts on behavioral patterns can have a therapeutic impact on children’s underlying biology as well, in conjunction with or beyond the provision of tangible resources. A plausible next step for inquiry is to examine whether PCIT’s effects extend beyond the behavioral gains documented in this clinical trial to determine if the improvements in children’s healthy eating-related behaviors are accompanied by measurable reductions in low-grade chronic inflammation and other biomarkers of metabolic health and immune function in high-risk child welfare-involved families. 

## 7. Future Research

Our findings suggest that PCIT appears to reduce child welfare-involved children’s higher-risk food approach behaviors (e.g., eating past satiety, emotional overeating, and speed of eating), though the intervention focused on strengthening the parent–child relationship rather than directly targeting eating behaviors themselves. Would the effects of PCIT on children’s eating behaviors be stronger if PCIT was adapted for use specifically during mealtimes? Can PCIT promote positive gains in children with chronic illnesses where eating concerns are prevalent (e.g., cystic fibrosis)? The results of this study suggest that PCIT is indeed an appropriate intervention for addressing higher-risk eating behaviors; however, randomized clinical trials of PCIT adapted specifically for mealtime interactions would extend the understanding of how PCIT can be implemented to specifically address problematic eating behaviors. Domhoff and Niec’s exciting new PCIT-Health adaptation shows promise for early obesity prevention and incorporates teaching parents specific mealtime skills that help to shape children’s healthy eating habits, reduce obesity-related feeding practices, and implement limits for screen usage [79]. Future clinical trial studies are needed to examine the relative benefits of PCIT-Health for child welfare-involved, ACE-exposed children who, although not overtly at risk for obesity, may evidence potentially concerning food approach behaviors stemming from living with various forms of instability (e.g., housing and food insecurity and parental insecurity due to child welfare involvement).

Moreover, additional research is needed to confirm if the effects of PCIT on child eating behaviors found in this study are unique to children exposed to toxic stress or if they are unique to families experiencing higher levels of food insecurity. As we did not specifically measure the levels of food insecurity in this study, replication with these additional measures is needed to clarify if and how food-insecure children benefit from PCIT relative to those who are food-secure. Whether PCIT’s beneficial effects on children’s eating behaviors are distinct to those experiencing food insecurity will be important to clarify the reach of PCIT in mitigating some of the negative impacts of living in under-resourced households. The current findings require further testing and replication in lower-risk, economically stable families as well, to understand the limits of PCIT’s effects on children’s eating behaviors. It may be that certain elements of PCIT are more effective for ACE-exposed families when addressing eating behaviors than for lower-risk families or, alternatively, that PCIT can be appropriate for families across differing demographic characteristics when targeting problematic eating behaviors. The findings from the current study open numerous avenues for continued investigation, and additional research is critically needed to pursue these important lines of inquiry. 

## 8. Study Limitations

Several limitations of this study are important to acknowledge. First, children’s eating behavior was assessed using a well-validated measure that was nonetheless based on parent-reporting, which may be subject to a response bias. Future clinical trials should include additional, objective measures of children’s food consumption (e.g., dietary recall and food logs), including, where possible, direct observations of child eating behaviors, in order to replicate and extend these findings. Second, long-term follow-up is needed to learn whether the observed improvements in children’s eating behavior can be sustained over time. Third, though the rates of engagement in the intervention were on par with other interventions with child welfare-involved families [80,81], this study did experience some attrition. Successfully addressing the barriers to engagement is crucial and an important avenue for further study. We examined attrition patterns within the larger clinical trial and documented behavioral, cognitive, and physiological differences in parents who dropped out of PCIT versus those who engaged [82], suggesting that understanding the treatment engagement and addressing potential barriers is possible and necessary to enhance treatment outcomes.

## 9. Conclusions

This study engaged a sample of child welfare-involved children and their parents who presented with limited economic resources and exposure to significant adversity, as evidenced by the majority of these primarily preschool children experiencing 3+ ACES upon study entry. The results provide new evidence for the beneficial effects of PCIT on children’s eating-related behaviors, with findings that can be expected to generalize to other child welfare-involved children with similarly high levels of economic and social adversity. 

## Figures and Tables

**Table 1 ijerph-19-10535-t001:** Demographic characteristics of children by the treatment condition (*n* = 204).

	Services as Usual Control (*n* = 84)	PCIT ITT (*n* = 120)	PCIT Engagers per Protocol (*n =* 79)
Child Race/Ethnicity			
European American/White	58.3%	56.7%	55.7%
Multiracial/Multiethnic	34.5%	38.3%	39.8%
Hispanic American/Latinx	3.6%	2.5%	3.4%
African American/Black	2.4%	0.8%	–
Asian/Pacific Islander	–	–	–
Native American/Alaskan Aleut	–	0.8%	–
Not reported	1.2%	0.8%	1.1%
Child age (years)	4.85 (1.44)	4.70 (1.48)	4.66 (1.34)
Child Sex			
Male	59.5%	51.7%	54.4%
Female	40.5%	48.3%	45.6%
Child ACES Exposures	3.65 (1.80)	3.29 (1.97)	3.63 (2.00)
Parent Sex			
Male	13.1%	10.8%	10.2%
Female	86.9%	89.2%	89.8%
Parent Education			
Less than high school	16.7%	16.7%	12.7%
High school	51.2%	48.3%	49.4%
Post-high school education	32.2%	34.0%	38.0%
Annual income (in dollars)	16,751.45	19,737.82	19,695.42

**Table 2 ijerph-19-10535-t002:** Effects of Parent–Child Interaction Therapy (PCIT) on children’s post-treatment eating behavior outcomes after accounting for the pre-treatment zcores.

	PCIT (Non-Imputed)		ITT				PCIT Engagers (Non-Imputed)		Per Prot				SAU Control (Non-Imputed)	
	Mean (SD)	Mean (SD)	b	SE	95% CI	*f* ^2^	Mean (SD)	Mean (SD)	b	SE	95% CI	*f* ^2^	Mean (SD)	Mean (SD)
	Pre-	Post-					Pre-	Post-					Pre-	Post-
Food Responsiveness	2.80 (0.90)	2.60 (0.84) *	−0.26	0.12	0.49, −0.02	0.03	2.94 (0.85)	2.61 (0.85) *	−0.28	0.13	0.55, −0.01	0.04	3.06 (0.95)	3.08 (1.04) *
Emotional overeating	1.79 (0.65)	1.68 (0.64) ^	−0.18	0.10	−0.38, 0.01	0.02	1.93 (0.72)	1.74 (0.68) ^	−0.19	0.11	−0.42, 0.03	0.03	1.92 (0.77)	1.96 (0.88) ^
Satiety responsiveness	2.90 (0.61)	2.96 (0.70)	0.05	0.10	−0.15, 0.24	0.00	2.93 (0.63)	3.05 (0.68)	0.13	0.10	−0.07, 0.34	0.02	2.78 (0.73)	2.80 (0.77)
Slowness in eating	2.85 (0.68)	3.07 (0.83) *	0.27	0.11	0.05, 0.49	0.04	2.98 (0.63)	3.15 (0.76) *	0.29	0.12	0.04, 0.53	0.05	2.97 (0.67)	2.89 (0.87) *
Emotional undereating	2.94 (0.83)	2.77 (0.97)	0.02	0.12	−0.22, 0.25	0.00	3.15 (0.76)	2.84 (0.91)	0.00	0.12	−0.24, 0.24	0.00	2.90 (0.72)	2.70 (0.70)
Food fussiness	3.11 (0.94)	3.28 (0.96)	0.05	0.11	−0.18, 0.27	0.00	3.15 (0.91)	3.34 (0.98)	0.08	0.13	−0.17, 0.32	0.01	2.85 (0.95)	3.00 (0.98)
Enjoyment of food	3.99 (0.77)	3.82 (0.82)	−0.09	0.10	−0.28, 0.11	0.00	3.97 (0.85)	3.77 (0.85)	−0.09	0.11	−0.31, 0.12	0.01	4.04 (0.77)	4.00 (0.84)
Desire to Drink	3.58 (1.02)	3.50 (1.02)	−0.11	0.14	−0.39, 0.17	0.00	3.76 (0.90)	3.52 (0.98)	−0.12	0.15	−0.42, 0.18	0.01	3.43 (0.88)	3.49 (1.05)

Note. Child Eating Behavior Questionnaire (CEBQ) scores. Higher scores indicating greater Food responsiveness, Emotional overeating, Satiety responsiveness, Slowness in eating, Emotional undereating, Fussiness, Enjoyment of food, or Desire to drink, respectively. Positive effect sizes represent greater gains for PCIT versus the control. * Significant mean differences at *p* < 0.05. ^ Mean differences at *p* < 0.10.

## Data Availability

Data available on request due to restrictions e.g., privacy or ethical. The data presented in this study are available on request from the corresponding author.

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
