# Peer review of "Parent–Child Interaction Therapy Supports Healthy Eating Behavior in Child Welfare-Involved Children"

_ijerph, 2022, doi:10.3390/ijerph191710535_

Round 1
Reviewer 1 Report
The authors report the results of an RCT comparing outcomes from a group receiving PCIT and a usual care control group. There are admirable features in the reported study: use of ITT, reasonable sample, etc. However, the writing style is repetitive/verbose, and substantial detail needs to be included. Also, changes in parenting practices/style (reported in another manuscript) need to be included here as possible mediating variables, to make a more coherent message (i.e. are changes in parenting style/practices related to changes in child eating behavior in expected/interpretable ways?)
The Introduction has repeated/redundant segments that should be condensed/deleted. Please get to the point. The Introduction could be usefully reduced by at least a third.
The writing style overall tends to be repetitive and should be correspondingly abbreviated (no redundancies).
Most of the literature on food parenting interventions shows no effect. Thus, the authors need to clearly identify problems in the existing literature and what they will do to minimize these problems in the hope of influencing adiposity.
If Domhoff et al. and Nice et al. have already employed PCIT to influence child eating behaviors, what did they find? And what are you doing that is innovative?
Parent report of healthy eating behaviors is a soft outcome subject to social desirability bias, especially among parents subjected to a parenting change intervention. Why didn’t the authors use a harder outcome, e.g. BMI, and test if changes in the parenting practices mediated change in BMI?
Why didn’t the authors screen on food parenting practices at recruitment to be sure each family employed parenting practices that needed to be changed?
Please provide a rationale for using a 1.5 to 1 intervention to control the allocation ratio, instead of 1 to 1.
The authors should also provide some assurance that the types of ACEs were comparable across groups.
Should screening for a minimal number of ACEs have also been employed?
At one point the authors report participants were recruited from Oregon DHS while at another point they state participants were referred by DHS. This requires a more detailed statement of recruitment procedures including who did the recruitment and who applied the screening criteria. What percent of the referred population did not agree? Were their demographic characteristics different from those in the sample? If Oregon DHS did the recruitment, how does the recruited sample differ from DHS enrollees to assess recruitment bias?
Over what time was the intervention delivered?
Were power calculations conducted? Why not?
Over what time were the families recruited? Did it include school year and summer break? Should a term for season be included in the stat model?
Why was there variability in the numbers of CDI and PDI sessions?
Who were the UOk coders? What training did they have in PCIT?
Please identify any possible overlap between PCIT and the services received by the control group.
Please provide Cronbach alphas for each of the CEBQ subscales in this sample.
What data interpolation procedures were employed?
The authors do not specifically identify “… other survey measure of family sociodemographics, child characteristics …” Since these may be related to the outcomes, they should be identified in the manuscript. Are any of these related to the outcome? Should they be controlled for in the analyses?
The primary target for intervention was parenting practices, but parenting practice effects are not reported in the manuscript. The authors are practicing salami slicing, i.e. maximizing the number of papers they can get out of a study. Analysis of the impact on parenting and child outcomes should have been reported together to assess if changes in the parenting practices mediated changes in child eating outcomes.
The detection of outcome effects in an RCT should be a treatment x time interaction term. It appears the table reports within-group differences since effects are reported for the controls. I do not see the results of an interaction term test in Table 2. P<0.10 is not meaningful in a large sample with many tests conducted.
Intention to treat analysis usually means that all randomized participants are included at post, not a specific test is not associated with it. What is ITT in Table 2?
The authors report a large number of tests. Should corrections be made for the number of tests or select a more conservative level for significance?
Was the intervention predicted to have effects on specific child outcomes, and not others? Otherwise, these are all exploratory analyses, which also suggests lowering the p-value.
The Abstract should more clearly state that this paper reports a secondary analysis on variables not specifically targeted by the intervention.
The discussion of obesity prevention should be minimized since the authors did not include BMI, and most family-based obesity prevention trials did not have the desired effects.
The authors discuss the effect of parenting style on child outcomes. Since the authors have already published the effect on parenting, the analysis in this paper should be revised/amended to include the possible mediation of parenting change on child outcome change.
Reviewer 2 Report
1. In the section of the introduction, I suggest the author could provide more information or consideration of the values of healthy eating behavior. Then the research values of child welfare-involved children with the development of healthy eating behavior would be provided. I think the core of this study should be focused on healthy eating behavior. The child welfare-involved children just is the studying sample.
2. Due to the lack of parenting engagement or involvement in healthy eating behavior of child welfare-involved children, the researcher wants to analyze their healthy eating behavior with the parenting intervention. I suggest the author could provide more information about this point.
3. In table 2 and related results, I suggest the author could reconsider the statistical analysis of these data and provide more information about this issue. The core of this study should be focused on the value of the intervention. So the statistical analysis should reveal the difference between pretest and posttest, and compare the differences among these children groups.
4. In the section of the discussion, I suggest the author could provide more information about the different values of the subscales with the intervention.
Round 2
Reviewer 1 Report
None.
Author Response
The reviewer had no additional comments.
Reviewer 2 Report
The author made more revisions to this manuscript based on my concerns. I suggest this manuscript should be accepted in the present edition.
Author Response
The author had no additional comments.